# ECOCAPTURE@HOME: Protocol for the Remote Assessment of Apathy and Its Everyday-Life Consequences

**DOI:** 10.3390/ijerph18157824

**Published:** 2021-07-23

**Authors:** Valérie Godefroy, Richard Levy, Arabella Bouzigues, Armelle Rametti-Lacroux, Raffaella Migliaccio, Bénédicte Batrancourt

**Affiliations:** 1Sorbonne Université, Institut du Cerveau—Paris Brain Institute—ICM, Inserm, CNRS, AP-HP, Hôpital de la Pitié Salpêtrière, F-75013 Paris, France; richard.dock2@gmail.com (R.L.); arabella.bouzigues.18@ucl.ac.uk (A.B.); armelle.ramettilacroux@icm-institute.org (A.R.-L.); lara.migliaccio@gmail.com (R.M.); benedicte.batrancourt@upmc.fr (B.B.); 2Département de Neurologie, Institut de la Mémoire et de la Maladie d’Alzheimer (IM2A), AP-HP, Hôpital de la Pitié Salpêtrière, F-75013 Paris, France; 3Unité de Neuropsychiatrie Comportementale, Département de Neurologie, AP-HP, Hôpital de la Pitié Salpêtrière, F-75013 Paris, France

**Keywords:** apathy, Alzheimer disease, fronto-temporal dementia, patient-caregiver dyads, remote monitoring, sensors, acceleration, electrodermal activity, blood volume pulse

## Abstract

Apathy, a common neuropsychiatric symptom associated with dementia, has a strong impact on patients’ and caregivers’ quality of life. However, it is still poorly understood and hard to define. The main objective of the ECOCAPTURE programme is to define a behavioural signature of apathy using an ecological approach. Within this program, ECOCAPTURE@HOME is an observational study which aims to validate a method based on new technologies for the remote monitoring of apathy in real life. For this study, we plan to recruit 60 couples: 20 patient-caregiver dyads in which patients suffer from behavioral variant Fronto-Temporal Dementia, 20 patient-caregiver dyads in which patients suffer from Alzheimer Disease and 20 healthy control couples. These dyads will be followed for 28 consecutive days via multi-sensor bracelets collecting passive data (acceleration, electrodermal activity, blood volume pulse). Active data will also be collected by questionnaires on a smartphone application. Using a pool of metrics extracted from these passive and active data, we will validate a measurement model for three behavioural markers of apathy (i.e., daytime activity, quality of sleep, and emotional arousal). The final purpose is to facilitate the follow-up and precise diagnosis of apathy, towards a personalised treatment of this condition within everyday life.

## 1. Introduction

Apathy is one of the most frequent neuropsychiatric symptoms within neurodegenerative diseases. It is often observed in individuals diagnosed with dementia, in particular in Alzheimer Disease (AD), Parkinson Disease (PD) [1], and Fronto-Temporal Dementia (FTD) [2]. Several studies have demonstrated its negative impact on the quality of life of both patients and their caregivers [3,4]. Apathy is indeed associated with a higher level of global functional impairment [5] and loss of autonomy in activities of daily living (ADL) [6,7]. Until now, pharmacological treatments of apathy have shown only moderate effectiveness [8] and among non-pharmacological interventions, only those that specifically adapt to each individual’s need bear potential for success [9]. Accurate assessment of apathy is of high importance in order to optimise the care management of behavioural disorders in neurodegenerative diseases. However, for various reasons, many obstacles have emerged in the process of evaluating the presence of apathy. These include the polymorphic nature of apathy, making it hard to define, and the limitations of current clinical assessment scales.

### 1.1. Assessment Limits and the Potential of Information and Communication Technologies

A first limitation regarding the assessment of apathy is due to its definition, which remains controversial. Robert Marin proposed that apathy corresponds to a ‘lack of motivation not attributable to diminished level of consciousness, cognitive impairment or emotional distress’ [10,11]. However, the concept of ‘lack of motivation’ is a projective psychological interpretation of a given behavioural state and as a syndrome, apathy should be objectively described, regardless of any psychological interpretation. This is the reason why apathy has more recently been defined, from a behavioural perspective, as the reduction of voluntary and purposeful behaviours [12,13]. Goal-directed behaviours (GDB) are observable and thus, more easily quantified. Moreover, apathy is usually assessed with clinical scales based on Marin‘s definition, with questions about internal state, thoughts and past activities suggesting a loss of motivation to perform daily activities (e.g., the Apathy Evaluation Scale [14]). These scales are thus biased by the subjective perspective of patients’ or caregivers’ point of view and they do not allow a precise description of apathetic behaviour in everyday life [15]. Besides, important discrepancies are often observed between patient’s and caregiver’s reports, especially in neurological conditions such as AD and FTD in which anosognosia causes a lack of awareness of symptoms [15].

Secondly, apathy is a complex multifaceted construct and the decrease in GDB may involve not only one mechanism (as suggested by Marin’s definition) but different underlying mechanisms: a disrupted emotional-affective processing, cognitive impairments and/or auto-activation deficits [12,15]. Indeed, Levy and Dubois [12,15] suggested a frontostriatal neurocircuitry model of three subtypes of apathy, apathy arising when any one of these processes is impaired. In this model, the emotional-affective aspects of apathy are assumed to be caused by impairments of the ventromedial prefrontal cortex (VMPFC)/orbitofrontal cortex (OFC) (connected to the limbic territories of the basal ganglia), cognitive impairments are supposed to be linked with damage in the dorsolateral prefrontal cortex (DLPFC) (connected to the cognitive territory of basal ganglia) and auto-activation deficits to impaired anterior cingulate cortex (ACC)/dorsal-medial PFC (connected to both cognitive and limbic basal ganglia territories). This model was supported by findings from Massimo et al. [16] in a population of individuals with behavioural variant frontotemporal dementia (bvFTD). This study linked motivation to the OFC, planning to the DLPFC, and initiation to the ACC. Interestingly, the profiles of apathy subtypes and their neural correlates seem to be disease-specific: in particular, emotional apathy is greater in bvFTD compared to AD [17,18] while executive apathy is greater in AD compared to bvFTD [17]. As suggested by Massimo and Evans [19], in a perspective of individualized-precision medicine, new tools are needed to objectively assess apathy as a polymorphic syndrome. Such tools would allow the precise assessment of the different mechanisms underlying apathy so that the individual’s specific troubles can be targeted. For this purpose, Massimo et al. [16] developed the Philadelphia Apathy Computerized Test which quantifies each of three components of GDB contributing to apathy through a computerized task based on reaction times under different conditions.

In accordance with the behavioural definition of apathy, it is possible to objectively assess apathy through markers derived from the direct tracking of one’s behaviour in every day-life using information and communication technologies (ICTs). ICTs are more and more used for telehealth to offer services that help to improve individuals’ health and well-being [20]. Telehealth services attenuate the morbid impact of chronic diseases due to a better and more regular/easier follow-up; they provide health care services without using hospital beds and they respond to the new needs of home care in an ageing population. The most remarkable advance in telehealth has been the development of telemonitoring to remotely monitor individuals’ behaviour in their real-life environments through the use of sensors providing so-called “passive” data (i.e., that do not require any active participation from the individual being monitored) [21]. Such an ecological approach allows to overcome several limitations that are inherent to evaluations in laboratory-like conditions: (1) the problem of test awareness, which may bias the subjects’ behaviour, (2) the lack of ecological validity, which limits the transfer of measures to more general settings, and (3) the impossibility to obtain continuous or frequent measures for monitoring purpose [21]. The volume of published studies using ICTs for the assessment of neurological conditions has thus increased exponentially from 1992 to 2017: very few studies were conducted before 2015, but more than 160 studies were listed on PubMed in 2017 [22]. ICTs, in particular smartphones and wearable sensors, bear great potential to improve the diagnosis and follow-up of apathy [13,23]. Using ICTs with an ecological approach could help the development of new tools providing a precise diagnosis of the specific apathy profile (in terms of underlying mechanisms). According to a group of experts working on brain disorders and apathy, ICTs could help the management of apathy, as they bear a great potential to improve remote personalized treatment [24]. Beyond the focus on patients’ neurological condition, these technologies may also contribute to the support of caregivers: they could help to reduce tension and stress in between patients and caregivers and to preserve caregivers’ psychological well-being [25].

### 1.2. Actigraphy and Its Limitations

To date, only actigraphy, which uses a piezoelectric accelerometer to measure acceleration on three orthogonal directions of space, has been proposed as an observer-independent and ecological approach to apathy evaluation. Ambulatory actigraphy with AD individuals showed that their apathy scores on neuropsychiatric assessment scales correlated negatively with mean motor activity measured by the actigraph [26,27]. Mullin et al. [28] also used ambulatory actigraphy to assess sleep-wake patterns in AD individuals. They observed that these patterns are also related to apathy scores on traditional clinical scales. In particular, among all AD individuals, those with apathy had significantly lower daytime mean motor activity, higher wake time after sleep onset, and higher total time in bed during the night than those without apathy. Using ambulatory actigraphy, Zeitzer et al. [29] also showed a higher decline in early afternoon motor activity along with an earlier wake and bed time in AD individuals with apathy compared to those without apathy. In another study by Valembois et al. [30], different activity patterns were observed between individuals with dementia and apathy and individuals with dementia and no apathy: those with apathy had a significantly lower level of motor activity between 9:00 a.m. and 12:00 p.m. and also between 18:00 p.m. and 21:00 p.m.

Although no study has investigated a long-term period (months) of actigraphic monitoring in individuals with apathy, one study [31] using actigraphy has followed individuals with dementia for a 2-week period. In addition, compared to previous ones, this study [31] presented the usefulness of assessing motor activity using actigraphy in patient-caregiver dyads instead of patients only. This research revealed significant differences in the level of motor activity comparing bvFTD individuals and their caregiver while these differences were not found to be significant between individuals with semantic dementia and their caregiver. More interestingly, lower levels of activity in individuals with bvFTD appeared to be associated with lower activity in the afternoon in their caregivers. BvFTD caregivers may find it too hard to engage an apathetic individual in activity, and since they have to monitor him/her, they might be unable to start an activity themselves. Another potential explanation is that bvFTD caregivers present a reduced activity related to their own emotional condition and lack of motivation facing another individual’s apathetic behaviour. The caregiver’s motor activity may therefore reflect and give further insight into the apathetic individual’s behaviour. This potential link between apathetic individuals’ level of motor activity and that of their caregiver’s should be further investigated.

Another limitation of actigraphic studies until now is that, while they provide an objective measure of motor activity through the use of scoring algorithms, details on the nature of the associated behaviours still cannot be inferred. For example, moments of high activity can be related to a goal-directed behaviour, but they may also be due to psychomotor agitation. Conversely, individuals may engage in a purposeful activity that is sedentary, such as talking on the telephone or working on a computer. It is thus necessary to examine additional data along with the analysis of actigraphic records to get more precise information on the behavioural consequences of apathy. Just as any mental health disease is difficult to diagnose on the basis of one or two questions about symptoms, it is very unlikely that a limited amount of metrics can accurately assess a complex neuropsychiatric syndrome such as apathy [32].

Overall, a better assessment of apathy in dementia requires the detection of relevant behavioural markers in the everyday life context, preferably in both individuals with dementia and their primary caregiver, using not only actigraphic measures of motor activity but also additional measures regarding the type of activity. This is the purpose of the research programme described hereafter.

### 1.3. The ECOCAPTURE Programme

ECOCAPTURE (see Figure 1), launched in 2014, is a research programme focused on the ecological and multi-modal assessment of apathy with a final goal of optimising the methods of diagnosis and therapeutic treatment of this syndrome. ECOCAPTURE is composed of two phases: Phase 1 corresponds to the first ECOCAPTURE clinical trial protocol (clinicaltrials.gov: NCT03272230), also called ECOCAPTURE@LAB, in which we explore the behavioural markers of apathy in a laboratory context reproducing a close-to-real-life situation (wait comfortably in a waiting room), using an acquisition system combining video and sensors (see Batrancourt et al. [33] for further details about this protocol); Phase 2 is developed through the ECOCAPTURE@HOME protocol (clinicaltrials.gov: NCT04865172) which is further described in this article.

ECOCAPTURE@HOME mainly focuses on the two following questions: Firstly, how can we remotely measure theoretical behavioural markers of apathy in a real-life context during a prolonged time period of four weeks? Secondly, is it possible to predict the psychological health status of a patient-caregiver dyad from the measure of these behavioural markers of apathy? The ECOCAPTURE@HOME project presents the originality to widen the concept of chronic disease beyond the patient’s state towards a systemic situation also involving his/her caregiver. Several studies showed that the role of a caregiver can have a very negative impact on their physical and/or psychological health (as well as on their financial and social life balance) [34], which in turn may affect their ability to carry out this role at the expense of the patient’s health. We therefore offer an original approach centred on both the individual with dementia and their spouse caregiver within a unitary system. This project should thus lay the foundations to create a telemonitoring system allowing the remote long-term follow-up of a patient’s status (in terms of apathy) along with its associated impact on caregivers’ well-being. Such a system could in particular be used to reach a higher level of precision in the diagnosis of apathy, through the definition of a patient’s specific apathy profile.

## 2. Materials and Methods

### 2.1. Objectives and Hypotheses of the Study

#### 2.1.1. Primary Objectives and Hypotheses

The main objective of ECOCAPTURE@HOME is to validate a measurement method to assess assumed behavioural markers of apathy in everyday-life conditions from data collected in patient-caregiver couples. We will attempt to identify the most fitted measurement model that ensures good psychometric properties (reliability and validity) for the measure of assumed behavioural markers of apathy. More precisely, we will validate two measurement models corresponding to two time-scales: across one month (28 days) and across one day. This will allow us to adjust according to the specific needs of clinicians regarding apathy telemonitoring: either short-term follow-up with daily measures or long-term follow-up with monthly measures.

Regarding this first central objective, we suggest three main hypotheses: the first one regarding the nature of assumed behavioural markers of apathy, the second one relative to the nature of the data required to assess these markers and the third one concerning the criteria used to guarantee their validity.

Firstly, we assume that the level of daytime activity, the quality of sleep and the emotional arousal can be three relevant behavioural markers of apathy in everyday-life conditions. Previous work using ambulatory actigraphy has shown that the intensity of daytime activity and the quality of sleep are relevant behavioural markers related to apathy [21,22,23,24,25,26]. We propose that daytime locomotor activity measured by actigraphy is an indirect marker of the apathy mechanism of auto-activation deficits, which reduces the individuals’ capacity to engage in activities at their own initiative. The relationship between sleep and apathy is most likely due to their common underlying physiological mechanism—that of being regulated by the circadian cycle [29]. We also hypothesise that a lack of emotional arousal can be a behavioural marker of apathy. Affective blunting is indeed a dimension of apathy assessed by clinical scales (in particular the Apathy Inventory by Robert et al. [1] and the Dimensional Apathy Scale by Radakovic et al. [35]) and functional neuroanatomy studies suggest that emotional processing is an underlying mechanism of apathy [12]. Moreover, emotional activation objectively measured by electroencephalography while viewing pleasant, neutral, and unpleasant pictures has been related to apathy: the centro-parietal late positive potential amplitude during unpleasant picture viewing was indeed more attenuated for individuals with PD reporting high apathy than for those with low or no apathy [36].

Secondly, we hypothesize that the combination of different types of data (passive and active) from diverse sources (patient and caregiver) will enhance the precision of the assessment of the behavioural markers of apathy. In particular, among passive data, not only acceleration data but also skin conductance and heart rate data should provide valuable information for the measure of the three behavioural markers of apathy. Metrics extracted from skin conductance [37] and heart rate [38] during sleep may be useful markers of the quality of sleep. Moreover, metrics quantifying the variability of skin conductance [39] and heart rate [40] have been related to emotional arousal.

Finally, we postulate that, in order to be valid, the three measured behavioural markers of apathy should: (1) distinguish patient-caregiver couples from healthy couples; (2) be able to predict the psychological health status of patient-caregiver dyads that is perceived by the caregiver (including the caregiver’s rating of the patient’s level of apathy) on a monthly and daily scale. This validation would provide a solid basis for the development of a tool for the remote follow-up of both individuals with dementia and their caregiver. Besides, we assume that it will be possible to disentangle different apathy profiles by data-driven clustering from the extracted scores on the behavioural markers of apathy. In particular, we should be able to identify distinct apathy profiles depending on the patient’s specific condition (i.e., AD or bvFTD).

#### 2.1.2. Secondary Objectives

To investigate the links between the extracted behavioural markers of apathy in real-life conditions and the self-reported measures of apathyTo investigate the impact of the caregiver’s perception of the dyad’s psychological health status on the subsequent behavioural markers of apathy.To investigate the relationships between the caregiver’s and patient’s respective passive behavioural data to test the hypothesis of a dynamic causal relationship between their physiological status.To show the capacity of passive behavioural data to predict active behavioural data, as a prerequisite for the future development of a machine learning system able to automatically infer active data from passive ones after a training period.

### 2.2. Population and Design of the Study

The ECOCAPTURE@HOME study is based on the recruitment of both patient-caregiver and control couples. Each dyad included in the study will be monitored through a multi-sensor wearable bracelet and questionnaires (on a smartphone application) for a period of 28 consecutive days. We plan to recruit a total of 60 couples between 40 and 85 years old divided into three groups: (1) Twenty patient-caregiver dyads with patients diagnosed with the behavioural variant of Fronto-Temporal Dementia (bvFTD), (2) Twenty patient-caregiver dyads with patients diagnosed with Alzheimer Disease (AD) and (3) Twenty healthy control dyads. Groups will be matched for age and socio-demographic characteristics.

The experimental design, summarised in Figure 2, is built upon the measurement model of the behavioural markers of apathy, presumed to predict the psychological status of the patient-caregiver dyad. Thus, we measure two types of variables on both a daily and a monthly scale:In all the dyads, the three theoretical behavioural markers of apathy: daytime activity level, quality of sleep, and emotional arousal, will be assessed using a pool of metrics extracted from the dyad’s raw passive and active behavioural data according to a hierarchical framework with three levels (raw data < metrics < markers considered as latent concepts);In patient-caregiver dyads only, the caregiver’s perception of the dyad’s psychological state is assessed by the following variables through questionnaires: (1) the patient’s level of apathy, as perceived by the caregiver; (2) the caregiver’s perceived burden; (3) the caregiver’s perceived quality of life.

### 2.3. Characteristics of Participants

#### 2.3.1. Selected Conditions

We chose to study two different neurodegenerative conditions (bvFTD and AD) in order to test the transnosological validity of our tool measuring the behavioural markers of apathy and their associated impact on the patient-caregiver dyad. The choice of these two conditions presents several advantages for our purpose: (1) caregivers of these patients are often stay-at-home caregivers; (2) these conditions do not involve any physical limitation (which could be a confounding factor in the assessment of activity level through acceleration measures); (3) since these two conditions are supposed to be related to different forms and severity levels of apathy [41], including both bvFTD and AD groups is likely to bring more variability within the measure of the behavioural markers of apathy. Moreover, bvFTD is an interesting model for apathy assessment since apathy is one of the major criteria for the clinical diagnosis of bvFTD [42] and is almost always present in this disease [2]. Apathy is also often observed in AD, though less often than in bvFTD, and it is one of the most frequently reported symptom at all stages of the disease [43].

#### 2.3.2. Recruitment and Inclusion Criteria

Patient-caregiver dyads will be recruited through neurological consultations by the neurologists of three neurological care units of the Salpêtrière hospital in Paris, France. Healthy control dyads will be selected using advertisements on dedicated websites.

Both patient-caregiver dyads and healthy control dyads should be couples living together at home either in a marital or non-marital status and both partners should own a smartphone. Moreover, individuals with dementia of patient-caregiver dyads should meet the following inclusion criteria:Diagnosis of bvFTD according to Rascovsky’s international criteria [42] for the bvFTD group/diagnosis of AD according to Dubois’s international criteria [44] for the AD group;No evidence of any other cerebral pathology;A Mini-Mental State Evaluation (MMSE) score superior or equal to 10 (to minimise the effect of confounding factors related to very severe cognitive impairment);Aged between 40 and 85;No evidence of any psychiatric condition and a Montgomery-Åsberg Depression Rating Scale (MADRS) score inferior to 20 (to avoid confusion between depression and apathy);No evidence of excessive consumption of psychotropic drugs—for instance benzodiazepines, sleeping pills, etc. (due to their tranquilising effect);No major physical disability disrupting mobility;No heart pacemaker (which would compromise heart rate measuring).

As for caregivers and partners of healthy control dyads, they must similarly comply with the following inclusion criteria:Aged between 40 and 85;No evidence of any psychiatric condition;A MADRS score inferior to 20;No evidence of excessive consumption of psychotropic drugs;No major physical disability disrupting mobility;No heart pacemaker.

### 2.4. Measure of the Behavioural Markers of Apathy

As shown in Figure 2, the behavioural markers of apathy will be assessed in both patient-caregiver dyads and healthy control dyads using behavioural metrics extracted from the dyad’s raw behavioural data. The raw behavioural data of a dyad is made of: (1) passive data, collected continuously, from the multi-sensor wearable bracelet in both members of the dyad; (2) active data, collected once a week during the four weeks of monitoring, from a questionnaire filled by one partner of the dyad (for patient-caregiver dyads, this will be the caregiver) using a smartphone application. For the monthly scale assessment, only the passive data throughout the 28 days will be used to extract the behavioural metrics whereas for the daily scale assessment, both the active and passive data throughout the 24 h will be used.

#### 2.4.1. Processing of Passive Behavioural Data from Sensors

Using validated algorithms, metrics will be extracted from the data of three sensors in both members of the dyad (for the daily and monthly scales): (1) metrics from the acceleration signal; (2) metrics from the skin conductance signal, and, (3) metrics from the Blood Volume Pulse (BVP) signal.

*Acceleration signal*: We will use a validated signal processing algorithm based on aggregation methods to extract «activity counts» from the raw data of acceleration. Using the validated Cole-Kripke algorithm (an automatic scoring method to distinguish sleep from wakefulness based on wrist actigraphy), we can also detect falling asleep and waking times from the acceleration signal.*Skin conductance signal*: Skin conductance data are traditionally interpreted using validated algorithms that extract two metrics: phasic skin conductance and tonic skin conductance. Phasic changes usually appear as abrupt increases (“peaks”) in the skin conductance. These are generally referred to as Galvanic Skin Responses (GSRs). Tonic skin conductance corresponds to the raw level of conductance of the skin and is usually referred to as Galvanic Skin Level (GSL). Further metrics such as storms (minimum of 5 GSRs/min for at least 10 consecutive minutes) can subsequently be derived from skin conductance data.*BVP signal*: The heart rate is computed by detecting peaks (beats) from the BVP data and computing the lengths of the intervals between adjacent beats. The inter-beat-interval (IBI) timing is used to estimate the instantaneous heart rate (HR), the average HR over multiple beats, and the heart rate variability (HRV) spectrum.

Aside from validated algorithms, we will also use functional principal component analysis (fPCA) to extract metrics from raw sensor data: strings of raw data for a given period can be fit with Fourier-based functions and the equations describing these fits are subjected to fPCA, which allows to reduce the complexity of the data sets through the extraction of only several components (see for instance Zeitzer et al. [29] using acceleration signals).

#### 2.4.2. Processing of Active Behavioural Data from Questionnaires

The questionnaire was completed once a week on a smartphone application and refers to five moments of the day: bedtime (3 items), wake time (3 items), breakfast (6 items), lunch (6 items), and dinner (6 items) (see Appendix A for further details on the item content of the questionnaire). In patient-caregiver dyads, these questions are completed by the caregiver who reports on the patient’s behaviour. Bed and wake time items will mostly allow to extract behavioural metrics related to the quality of sleep whereas mealtime items are used for the extraction of metrics associated with daytime activity and emotional arousal.

#### 2.4.3. Metrics for Daytime Activity, Quality of Sleep, and Emotional Arousal

*Daytime activity*: After extracting the activity counts and distinguishing periods of wake and sleep from the acceleration signal, it is possible to calculate more elaborated metrics such as the mean number of activity counts per minute during daytime. We will also detect bouts of activity (sustained periods of elevated counts), bouts of sedentary time (sustained periods of low counts) and compute their mean frequency and duration during daytime. Using active data, we will assess the perceived partner’s investment in several activities at mealtimes: cooking, cleaning, washing the dishes, etc., which will also be useful as complementary daytime activity metrics.

*Quality of sleep*: After extracting the activity counts and distinguishing periods of wake and sleep from acceleration signal, metrics such as total sleep time, wake time after sleep onset, frequency, and duration of activity bouts during night-time will be extracted. We will also integrate metrics extracted from skin conductance and BVP signals during sleep as potential markers of the quality of sleep. The active data enable the calculation of additional metrics such as total time in bed (time between reported bed and wake times), sleep latency (time between reported bedtime and estimated sleep onset), difficulty to wake up and signs of fatigue for one of the dyad partners (patient in patient-caregiver dyads).

*Emotional arousal*: We will use metrics extracted from skin conductance and BVP signals. Metrics derived from the phasic parameter of skin conductance (GSRs) will enable the evaluation of emotional responses to short-term events (e.g., environmental stimulations). Metrics related to the variability of HR (HRV spectrum) are also considered as measures of emotional arousal. Active data will provide complementary metrics such as the perceived partner’s emotional response during interactions at mealtimes.

### 2.5. Measure of Caregiver’s Perception of the Dyad’s Psychological State

As described in Figure 2, we assess the dyad’s psychological status only in patient-caregiver dyads from the caregiver’s point of view. Similar to the behavioural markers of apathy, the caregiver’s perception will be measured on two scales: (1) on a monthly scale, through three validated questionnaires; (2) on a daily scale, through three Visual Analogue Scales (VAS).

On the monthly scale, three questionnaires will be completed at the end of the 28-day monitoring to assess the caregiver’s perception of the dyad’s psychological state for these 28 days. The caregiver will thus complete: the Apathy Scale (AS) [45] (informant-rated version) to rate his/her perception of patient’s apathy level, the Zarit Burden Interview (ZBI) (12-item version validated by Bédard et al. [46]) to rate his/her perceived burden, and the Short Form 36 (SF36) [47] to assess his/her perceived health-related quality of life (i.e., physical functioning, social functioning, role limitations due to physical problems, role limitations due to emotional problems, mental health, energy and vitality, pain, general perception of health, and changes in respondent’s health over the past year).

On the daily scale, the caregivers will complete only three VAS once a week at the end of the day to assess the three variables related to their perception of the dyad’s psychological state (the patient’s apathy, associated burden, and health-related quality of life). Validated questionnaires are often time-consuming, hard to complete and therefore not suitable for daily assessment tools. Only one item with a Visual Analogue Scale (VAS) can be used as a reliable substitute for a whole questionnaire on a daily basis, in particular for the assessment of perceived quality of life [48].

### 2.6. Procedure

#### 2.6.1. Visit 1 and Visit 2

At visit 1 (see Table 1), the recruited dyads come to the Paris Brain Institute (ICM, Paris, France) at the Centre d’Investigation Clinique Neurosciences (CIC Neurosciences), located at the Salpêtrière hospital in order to: (1) complete the inclusion process; (2) undertake several tests to assess the severity of dementia and of specific clinical symptoms (related to the study); (3) receive all the material for the remote follow-up and be trained to use it. At visit 1, we assess self-reported apathy using two complementary questionnaires: the Apathy Scale (AS) [45] and the Dimensional Apathy Scale (DAS) [35]. The AS is a clinical scale assessing apathy defined as a unidimensional construct linked with general lack of motivation [45] whereas the DAS measures three different underlying dimensions of apathy [35]. Indeed, the DAS consists of three subscales respectively measuring the Emotional, Initiation, and Executive subtypes of apathy inspired by the theoretical model of Levy and Dubois [12,15]. These questionnaires will be useful to investigate the relationships between the behavioural markers of apathy (calculated from passive and active data collected during the remote follow-up—see Section 2.4.1, Section 2.4.2 and Section 2.4.3) and self-reported apathy.

Visit 2 at the Paris Brain Institute (ICM) occurs to return the material and for a feedback session with the couple (perceived benefits, burden of the protocol, potential improvements).

#### 2.6.2. Planning of Remote 28-Day Follow-Up

The remote follow-up of a couple lasts for 28 days during which partners should adopt their usual way of life. The follow-up starts at T0, on the morning of the first Monday after visit 1 and ends at T0 + 28 days, on the morning of the fourth Monday after T0 (see Figure 3). Visit 2 is scheduled as soon as possible from T0 + 28 days at the Paris Brain Institute.

As shown in Figure 3, we collect two types of data: (1) passive behavioural data from sensors, nearly continuously during the four weeks of follow-up; (2) active data from online questionnaires, once a week on a day chosen by the partner who fills the questionnaires (active behavioural data and in patient-caregiver dyads, active data on dyad’s psychological status on a daily scale) and at T0 + 28 days (in patient-caregiver dyads, dyad’s psychological status on a monthly scale).

To retain and support couples included in the study during the four weeks of follow-up, we will schedule two appointments on the phone with an investigator. These appointments will not only allow the investigator to check on the successful completion of the follow-up and to answer the couple’s questions (about technical issues for instance), but they will also provide social support for caregivers who may sometimes feel isolated. Besides, couples included in the study will have the possibility to contact an investigator on business days if needed (see the Instruction and information brochure in Appendix A). To further ensure a good quality of data collection, we will reward proper compliance with the protocol through financial compensation (€100 per couple).

#### 2.6.3. Wearable Sensor System

Both members of the recruited dyads will wear, on the non-dominant wrist, the E4 wristband (Empatica Inc., 1 Broadway, Cambridge, MA, USA) which is a wearable research device that offers real-time physiological data acquisition through four sensors:A 3-axis accelerometer, which measures acceleration on three orthogonal spatial directions;A Galvanic Skin Response (GSR) sensor, which measures the electrical conductance of the skin;A photoplethysmography (PPG) sensor, which continuously measures Blood Volume Pulse (BVP, i.e., volumetric variations of blood circulation) using a light source and a photodetector at the surface of skin;A thermopile infrared sensor, which reads peripheral skin temperature.

Besides the four sensors, the E4 wristband is equipped with an event mark button, which can be used to tag events and link them to the recorded physiological signals. We will ask one member of each dyad (caregiver in patient-caregiver dyads and RCM in healthy control dyads) to use this button to mark events with supposed strong emotional charge for their partner. This information can indeed be useful to interpret and check the validity of the GSR and PPG signals.

The main instructions for the use of the E4 wristband (see Appendix A for more details on the instructions given to patient-caregiver dyads) are as follows: (1) wear the bracelets on the non-dominant wrist as often as possible during the daytime and nightime; (2) keep the smartphones with you as often as possible (for data retrieval from E4 bracelets to smartphones); (3) beware of avoiding the exchange of your bracelet with your partner’s; (4) charge the battery of E4 bracelets every evening (approximative time of charge: one hour and a half).

#### 2.6.4. Questionnaire Interfaces on Smartphone Application

To collect all the active data from the questionnaires, a smartphone application (sending notifications) will be used by the caregiver (in patient-caregiver dyads). On this application, there will be two interfaces:−The day questionnaire interface will be used by one partner (caregiver in patient-caregiver dyads) one day per week to provide information on their partner’s behaviour at: (1) Bedtime (the day before); (2) Wake time; (3) Breakfast; (4) Lunch; (5) Dinner. This interface will also be used at the end of the day by the caregiver (only in patient-caregiver dyads) to provide their perception of the dyad’s psychological status (i.e., patient’s apathy, perceived burden and quality of life) during the day.−The month questionnaire interface will be used by the caregiver (only in patient-caregiver dyads) at T0 + 28 days to complete the three validated questionnaires (AS, ZBI, SF-36) on their global perception of the dyad’s psychological status (i.e., patient’s apathy, perceived burden, and quality of life) during the 28 days of follow-up.

### 2.7. Data Flow

All the information collected during Visit 1 will be registered using REDCap^®^ (version 10.9.4) secure web application. These data will be stored on the local secure server (of Paris Brain Institute, Paris, France).

During the 28-day follow-up, remote data collection will require two steps (see Figure 4):Collection of two types of data through an application installed (by the investigators) on patients’ and caregivers’ smartphones: data from sensors (passive behavioural data), and data from questionnaires (active behavioural data + caregiver’s perception of the dyad’s psychological state);Transfer of these data to a remote server application hosted by a local secure server (of the Paris Brain Institute).

Data from the sensors of the E4 bracelet will be continuously collected in both partners of the dyad and transferred to their smartphone application in real-time, each bracelet being paired with its carrier’s smartphone. In case of connection loss with the paired bracelet, the application will send a push notification to warn the participant. Data from questionnaires will be collected directly through the application, sending reminder notifications to the caregiver in patient-caregiver dyads (or to one of the partners, randomly selected, in control dyads).

After being encrypted, all the data collected by the application will be temporarily stocked on the smartphone and progressively transferred to the remote server application hosted by the local secure server. All the data collected by the application will be anonymised and identified only through the associated time-stamp and individual PIN code attributed to each participant of the study.

### 2.8. Statistical Analyses

#### 2.8.1. Sample Size

The required sample size is estimated according to our primary objective of validation of a measurement model for the three assumed behavioural markers of apathy. We will use a Confirmatory Factor Analysis (CFA) based on Structural Equation Modelling and according to Westland [49]. For this kind of models, the required sample size depends on the ratio r = p/k where p is the number of observed indicator variables (i.e., the total number of metrics) and k is the number of unobserved latent variables in the model (i.e., the number of behavioural markers of apathy we want to measure through these metrics). Assuming that we can find 36 metrics for the measure of the three behavioural markers of apathy for one dyad, a ratio r = p/k = 36/3 = 12 would require a sample size of at least 50. We thus decided to recruit a total of 60 dyads, which is 20% more than this minimum sample size, to compensate for potential dropout of some dyads during follow-up. Besides, a total of 60 dyads seems to be a suitable sample size in terms of feasibility (given that we plan to complete the inclusions within a maximum of 24 months by recruiting 2 or 3 dyads per month). With this sample size (N = 60) for a model with p = 36 and k = 3 (591 degrees of freedom), the statistical power (i.e., the probability of rejecting the null hypothesis of a model with close fit when true model fit is mediocre) is quite satisfactory (π = 94.2%).

#### 2.8.2. Analysis Plan for Primary Objective

First, we will extract the metrics from both passive and active behavioural data on a monthly and a daily basis for all dyads. Using these metrics, we will explore a measurement model for the three assumed behavioural markers of apathy on the two time-scales (with monthly scale metrics for the monthly scale model and daily scale metrics for the daily scale model).

We will use Exploratory Factor Analysis (EFA) and Confirmatory Factor Analysis (CFA) in order to identify the system of metrics organised between the three markers, which is the most adapted to measure them. Statistical criteria for the validation of the measurement model tested in CFA (i.e., Root Mean Square Error of Approximation < 0.08, Standardized Root Mean Square Residual < 0.08, Comparative Fit Index > 0.9, Tucker-Lewis Index > 0.9) will guarantee its good adjustment to the data. The internal consistency of the measures of the three behavioural markers will be assessed using Cronbach’s alpha (>0.70). The reliability of the measures (on a daily scale) will also be evaluated through the correlations between the behavioural marker scores obtained on four time-steps (once a week) during the 28-day follow-up (high correlation indicating high reliability).

We will compare the measurement models and mean scores on the three behavioural markers of apathy between groups of patient-caregiver dyads and control dyads (on two time-scales). We will demonstrate that the patient-caregiver dyad’s scores on the three assumed markers of apathy can predict the dyad’s psychological state (as perceived by the spouse caregiver) using linear regression models on the monthly and daily scales. In a perspective of precision medicine, we will also investigate the disentangling of different profiles of apathy, using clustering analyses from the calculated behavioural markers of apathy.

#### 2.8.3. Analysis Plan for Secondary Objectives

A.We will investigate the links between the extracted behavioural markers of apathy in real-life conditions and the self-reported measures of apathy using a linear regression. We may also compare the clusters of individuals with dementia identified based on the behavioural markers with those based on the self-reported measures of apathy (DAS subscales in particular).B.We will analyse the impact of caregiver’s perception of the dyad’s psychological status on the subsequent behavioural markers of apathy using a linear regression to show that the caregiver’s perception at day N predicts the three behavioural markers of apathy at day N + 1 (i.e., one week later) above and beyond the markers of apathy at day N.C.We also will study the relationships between the patient’s and his/her caregiver’s passive behavioural data using Granger causality test (to investigate causality between two variables in a time series) applied to the different sensor signals collected for 28 days in patient-caregiver dyads.D.We will attempt to show that passive behavioural data can predict active behavioural data collected on a daily scale. For this purpose, we will create categories derived from the active behavioural data (for instance, good–medium–bad day from the caregiver’s point of view) and use different supervised clustering methods (Linear Discriminant Analysis, Random Forest, Support Vector Machine, Artificial Neural Networks, K-Nearest Neighbours) to investigate their capacity to predict the correct category from the set of passive behavioural data.

## 3. Expected Results and Discussion

### 3.1. Expected Results

We expect to identify a fitted measurement model for daytime activity, quality of sleep and emotional arousal from the collected passive and active data in all dyads. This will be the starting point to obtain scores for these three behavioural markers of apathy in each dyad (four scores on a daily scale and one score on a monthly scale). Scores of daytime activity, quality of sleep and emotional arousal are expected to be highly correlated and to distinguish patient-caregiver couples from healthy control couples. Besides, these scores should all contribute to predict the caregiver’s perception regarding the patient’s apathy and regarding his/her own burden and quality of life. Finally, we expect to identify different apathy profiles based on the extracted scores of daytime activity, quality of sleep, and emotional arousal. In particular, as emotional apathy has been shown to be greater in bvFTD than in AD [17,18], we should observe a profile characterized by very low emotional arousal in bvFTD.

Regarding the secondary objectives, we expect to be able to identify which behavioural marker (daytime activity, quality of sleep, or emotional arousal) best predicts the different self-reported measures of apathy. In particular, we can expect the initiation (or auto-activation) component of apathy (DAS-Initiation) to be specifically related to daytime activity whereas the emotional apathy subtype (DAS-Emotional) should be more linked with emotional arousal. Moreover, we globally expect to provide evidence of a close and reciprocal relationship between the patient’s behaviour/status and caregiver’s behaviour/status (e.g., strong causal impact between patient’s and caregiver’s physiological status assessed by sensor data). The caregiver’s state should have an impact on the subsequent measure of daytime activity, quality of sleep, and emotional arousal and reciprocally.

### 3.2. Potential Limitations of the Project

The first global limitation of the study is the risk of high dropout and poor compliance with the protocol in the recruited dyads, especially in patient-caregiver dyads who have to deal with everyday issues related to the patient’s functional difficulties. Solutions will be implemented to maintain motivation in order to limit dropout and poor compliance. Firstly, a support by phone will be offered during the follow-up (two scheduled appointments on the phone with an investigator and possibility to contact him/her if needed). This will constitute an opportunity for the investigator to warn participants in case of poor compliance (the investigator can follow data collection for each couple during follow-up thanks to a dedicated dashboard). Secondly, the financial compensation for taking part in the study will depend on proper compliance with the protocol of the whole 28-day follow-up. Besides, the protocol has been optimized to limit missing data due to poor compliance. Regarding the compliance with the wearing of the sensor bracelet, as participants are supposed to wear it nearly continuously during the follow-up, it should be less likely for them to forget it. Moreover, to avoid technical issues preventing sensor data collection, warning notifications will be sent by the smartphone application when the bracelet battery is low and as soon as the bracelet is disconnected from the app. It may be hard for some individuals with dementia to remember that they have to keep the bracelet on, but we expect that caregivers will help them with bracelet wearing. Regarding the compliance with the filling of questionnaires, notifications will be sent by the application to remind to complete them, and participants will be forced to answer all the questions to be able to send their report. Some questionnaires (those expected at T0 + 28 days) could also be completed at Visit 2 if it has not been done before. In case of missing data due to non-compliance with bracelet wearing and/or questionnaire filling during follow-up, we will try to use the available information as much as possible to contribute to the validation of measurement models on a daily and monthly scale (as long as at least 80% of the total amount of required data has been collected).

Several issues can also be discussed regarding measurement bias in the assessment of the behavioural markers of apathy from passive and active behavioural data. Measurement bias may firstly appear in passive data from sensors because they depend on many parameters that we cannot always control. In particular, measures of acceleration and GSR depend on the placement of the sensor on the body. GSR assessment has indeed been shown to differ significantly across the two halves of the upper body and traditional measures on only one side may lead to misjudgement of arousal [50]. The placement of the sensors on only one wrist in this protocol is therefore a potential issue. However, as wrist-worn devices are common, they are well accepted and individuals generally have no objection to wearing a bracelet or watch, which will increase involvement and compliance with the study [21]. Moreover, GSRs and their different phase peaks are very dependent on contextual variables that can be difficult to control. Indeed, in everyday-life conditions, it is difficult to keep a precise record of the situations in which the emotional reactions occur. This is the reason why skin conductance is generally measured under laboratory-like conditions. However, recent studies (e.g., Can et al. [51]) used GSR sensors (combined with other sensors) in real-life environments and showed that measures of skin conductance contributed to reach reliable classifications of stress level by machine learning algorithms. Regarding the active behavioural data provided by the caregiver once a week (to complete the measure of apathy markers), it is possible that the frequent monitoring of a patient’s behaviour will lead to more awareness of everyday difficulties in caregivers, thus potentially biasing their assessment of the patient’s behaviour.

Another potential measurement bias can be underlined in the assessment of behavioural markers of apathy through metrics from passive data of both individuals with dementia and their caregiver. Residual errors of prediction for the passive behavioural metrics aimed at measuring one behavioural marker will not be independent from each other and identically distributed: in particular, the residual errors relative to the caregivers’ metrics are likely to be covariates and so are the residual errors relative to the patients’ metrics. These covariances will have to be taken into account in the statistical modelling. It is also possible that the metrics extracted for both partners of one dyad are not sufficiently correlated to correspond to the measure of the same latent behavioural marker (especially for patient-caregiver dyads). If so, behavioural markers of apathy will need to be assessed using only the patient’s passive behavioural data instead of both the patient’s and caregiver’s passive behavioural data.

Finally, it seems difficult to estimate the extent of the measurement bias that might be introduced by the support provided to the caregiver through the study follow-up in an objective way. We could try to investigate a potential modification of signals from sensors worn by the caregiver between before and after a phone call with the investigator. However, we expect that this impact can be neglected (only two phone calls are scheduled with the investigator, and this is mostly for technical support).

### 3.3. Potential Impacts of the Project

The aim of this protocol is to lay the foundations for a system capable of remotely monitoring the evolution of apathy in individuals with dementia and the associated impact on their caregiver. This system would provide a measure of behavioural markers of apathy that is completely independent of the patient’s self-report and thus not impacted by the patient’s subjective bias and anosognosia. Building on the knowledge acquired through this first ECOCAPTURE@HOME study, we expect that the following step of such a process will be to test a machine learning system which could, after a training period, automatically estimate the behavioural markers of apathy and the associated caregiver’s perception of the dyad’s status, using solely passive data from sensors. This system could present several interesting applications for clinicians and in particular neurologists. Firstly, this monitoring system could allow for the remote evaluation of the evolution of the patient-caregiver dyads’ psychological state, thus improving the potential for detection and prevention of dangerous situations such as caregiver burn out. Moreover, it could be used as a diagnostic tool providing information about a patient’s specific profile on the different behavioural markers of apathy and on how it is related with caregiver’s psychological status. This could prove useful for the implementation of new treatments adapted to a couple’s specific characteristics. In particular, by enabling the daily assessment of behavioural markers of apathy and associated dyad’s psychological health status, this system could be of great use to test and adapt therapeutic interventions accordingly in patients’ homes.

## 4. Conclusions

The ECOCAPTURE@HOME project focuses on the assessment of behavioural markers of apathy with an ecological approach in the everyday life context. It is characterised by three main innovative aspects: (1) the multi-modality of the assessment involving both passive and active data, with passive data combining measures from three sensors; (2) the focus on patient-caregiver dyads as a unitary system (instead of focusing exclusively on patients); (3) the possibility for concrete applications through the creation of a telemonitoring system allowing the remote follow-up of patient-caregiver couples and the identification of their specific characteristics and needs. The development of home-based therapeutic interventions targeting apathy could be highly facilitated using such a system.

## Figures and Tables

**Figure 1 ijerph-18-07824-f001:**
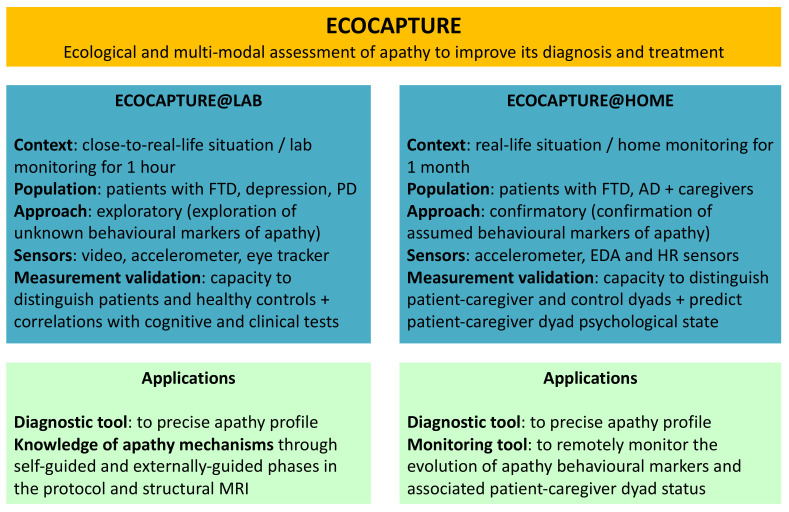
Methodological characteristics and applications of ECOCAPTURE studies. AD: Alzheimer’s disease; FTD: frontotemporal dementia (behavioural variant); PD: Parkinson’s disease; EDA: electrodermal activity; HR: heart rate.

**Figure 2 ijerph-18-07824-f002:**
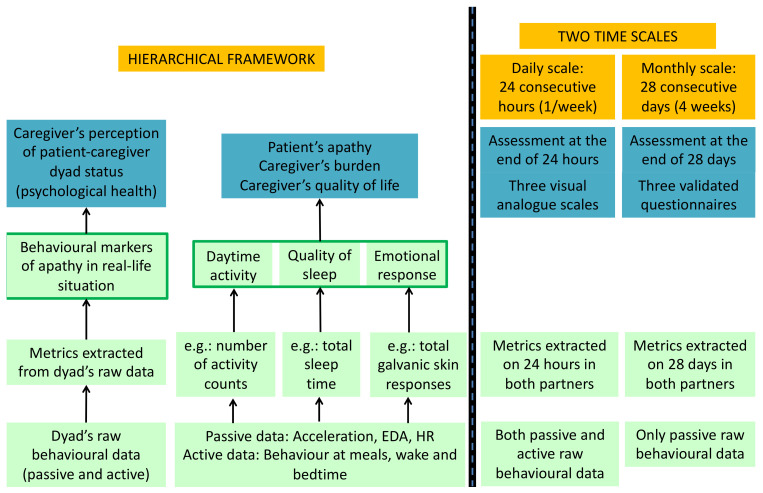
Experimental design with two broad kinds of variables: (1) the behavioural markers of apathy (in green); (2) the caregiver’s perception of patient-caregiver status (in blue). EDA: electrodermal activity (or skin conductance); HR: heart rate.

**Figure 3 ijerph-18-07824-f003:**
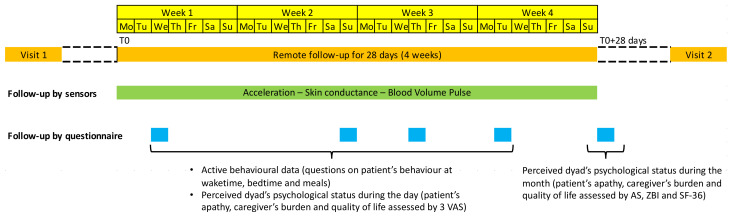
Example of follow-up calendar for a patient-caregiver dyad. Sensors are worn by both patient and caregiver. Questionnaires are completed exclusively by the caregiver.

**Figure 4 ijerph-18-07824-f004:**
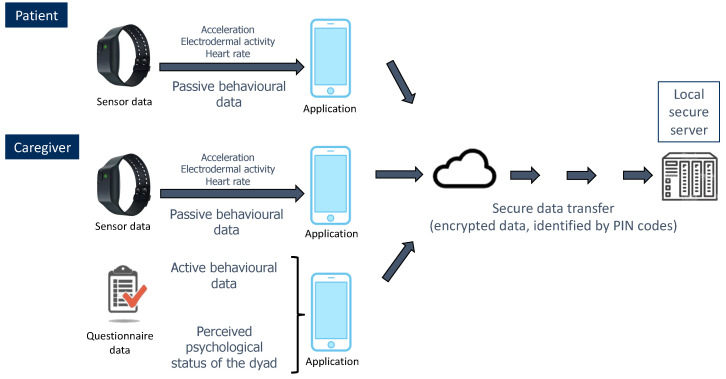
Summarised data flow during the 28-day follow-up.

**Table 1 ijerph-18-07824-t001:** Detailed description of visit 1.

Inclusion (35 min)	
5 min	After giving further information to the recruited couple on the objectives/conditions of the protocol and making sure that all is clearly understood, the investigator gives to both partners an information notice and collects their non-opposition to participate in the study. If a patient has a legal guardian, non-opposition is provided by the guardian and if a patient is under curatorship, non-opposition is provided by the patient with the help of their curator.
5 min	For all the participants, the collection of demographics (age, sex, and education level), medical history (in particular, date of first symptoms and diagnosis for patients) and treatments.
25 min	Verification of inclusion criteria including two tests: − For patients, MMSE: general cognitive efficiency − For all the participants, MADRS: depression symptoms.
**Assessment of the Severity of Dementia and of Clinical Symptoms of Interest (1 h)**	
30 min	For patients, Clinical Dementia Rating (CDR): severity of dementia symptoms in three domains (cognition, autonomy, behaviour).
30 min	For all the participants, − Apathy Scale (AS, self-reported version) − Dimensional Apathy Scale (DAS) − Epworth Sleepiness Scale (ESS).
**Handing over of the Equipment and Training (15 min)**	
15 min	Handing over two bracelets per couple; installation and configuration of a smartphone application for each partner for remote data collection; Training for the use of bracelets and application; for control dyads, random selection of the partner who fills the questionnaire (for active behavioural data) once a week; answering potential questions; Choice of a date for the first telephone conversation with the investigator during the 4-week remote follow-up.

## Data Availability

Only study members at the Paris Brain Institute will have access to the full dataset once study data has been collected. Results will be disseminated via publications in peer-reviewed journals. There are no plans to grant the public access to the dataset or statistical code.

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
