# Peer review of "ECOCAPTURE@HOME: Protocol for the Remote Assessment of Apathy and Its Everyday-Life Consequences"

_ijerph, 2021, doi:10.3390/ijerph18157824_

Round 1

Reviewer 1 Report

The article presents a protocol for remote monitoring of patients with dementia (subclinical diagnosis of Alzhiemer's and Fronto-Temporal) to monitor their symptoms of apathy through ICT technology. 

The article is a great contribution, necessary and very relevant, with a strong technical and theoretical support. I would highlight two issues that give intrinsic value to this article. On the one hand, the theoretical perspective, basically behavioral and moving away from unobservable inferences of psychic phenomena. On the other hand, the contribution of a monitoring protocol applying new technologies and its application to clinical or sub-clinical groups. Undoubtedly it is fundamental given the high development that wireless technology brings us in physiological records. Furthermore, this protocol and its publication is fundamental for subsequent adjustments and readjustments to new monitoring protocols, whose contribution has a high intrinsic value independently of the clinical results.

I would like to underline that GSRs (Galvanik Skin Responses) and their different phase peaks are very dependent on contextual variables. It is a very good measure in the laboratory setting, since it is very sensitive to contextual and emotional variations. However, in natural and ecological contexts, without keeping a strict record of the concrete situation in which the reactions occur, I maintain certain reservations as to its real value. 

Congratulations to the authors for their initiative and development.

Reviewer 2 Report

This is an interesting project and this protocol article provides a comprehensive and accessible summary.

Line 103: I think this should be 12pm rather than 12am.

Use of the word 'patient' to describe people with dementia, particularly in paragraph starting on line 106. Patients are people actively in receipt of healthcare, so if they are at home they are people with dementia in the same way that when I'm at home I'm just a person not a patient. I can see that participants are being recruited from clinics but even still would suggest revision throughout the paper.

line 234 - use of the word 'suffering' - would recommend changing to diagnosed with or living with as people with dementia don't typically acknowledge suffering as a descriptor for themselves and it is a stigmatising word. Check throughout.

Reviewer 3 Report

Thank you for submission of the manuscript. It outlines a study protocol exploring surrogates measures for apathy in behavioural variant Frontotemporal Dementia (bvFTD) and Alzheimer’s Disease (AD). This is a comprehensive protocol, with interesting focus. Please see below comments/suggestions/queries:

  1. In the Introduction section, it would be useful if the authors expanded more on previous research on multidimensional apathy in AD and bvFTD, to highlight this within the context.
  2. In the Introduction section, the authors may find the below reference relevant as it discusses ICT relative to managing apathy:
  3. In the Introduction and Expected Results and Discussion section, the manuscript would benefit from further discussion in relation to insight or awareness problems that are characteristic of bvFTD (as well as AD) and how this might impact the findings and results relative to the self-rated measures.
  4. The Materials and methods section, for the objectives, the aims and hypothesis are not really clear. Perhaps the authors could consider highlighting the overarching aims and hypotheses at the beginning of each section? This would be helpful, prior to going in to in depth exploration of the aims and hypotheses.
  5. Could the authors justify why two Apathy scales are being used in the protocol?
  6. Will the authors be examining associations of different Dimensional Apathy Scale subscales and various remote monitoring measures? Please clarify.
  7. Are the author’s able to explain why and informant/caregiver-rated measure of apathy was not included?
  8. For Figure 3, it is not clear what questionnaires will be completed. Are they the same as those administered in Visit 1? Please clarify further.
  9. In the Sample size section, the authors state that they will recruit “20% more than this minimum sample size”. How is this justified? Attrition or dropout might be higher for this study. Are there any methods in place to try to counter attrition or dropout (i.e. retention)? Please discuss further.
  10. Further, more generally and in reference to the Potential Limitation subsection, how will the author account for missingness of data in terms of non-compliance (e.g. forgetting to put bracelets on) or non-completion (relative to questionnaire measures)?
  11. General question, how will the author’s account for caregiver support provided to people in the study? Might it be possible that the level of daytime activity and emotional arousal might be mediates by caregivers? Please clarify and perhaps discuss.

Round 2

Reviewer 3 Report

Thank you to the authors for attending to all the points raised by this reviewer. These were comprehensive and helpful in improving the manuscript further. Apologies for not adding a suggested reference relating to the ICT relative to managing apathy point, this was done in error. Please see below reference:

Manera, V., Abrahams, S., Agüera-Ortiz, L., Bremond, F., David, R., Fairchild, K., ... & Robert, P. (2020). Recommendations for the nonpharmacological treatment of apathy in brain disorders. The American Journal of Geriatric Psychiatry, 28(4), 410-420.

Additionally, the authors may wish to consider the below apathy subtype and bvFTD literature, for Introduction and Expected Results and Discussion sections, if it is relevant to the manuscript:

Wei, G., Irish, M., Hodges, J. R., Piguet, O., & Kumfor, F. (2019). Disease-specific profiles of apathy in Alzheimer’s disease and behavioural-variant frontotemporal dementia differ across the disease course. Journal of neurology, 1-11.

Radakovic, R., Colville, S., Cranley, D., Starr, J. M., Pal, S., & Abrahams, S. (2020). Multidimensional apathy in behavioral variant frontotemporal dementia, primary progressive aphasia, and Alzheimer disease. Journal of geriatric psychiatry and neurology, 0891988720924716.
